# Study to evaluate the readability and visual appearance of online resources for blunt chest trauma: an evaluation of online resources using mixed methods

Hayley Anne Hutchings ![ORCID],[1] Max Cochrane,[1] Ceri Battle ![ORCID] [1,2]

¹School of Medicine, Swansea University, Swansea, UK
²Physiotherapy Department, Swansea Bay University Health Board, Swansea, UK

**Correspondence to**
Professor Hayley Anne Hutchings;
h.a.hutchings@swansea.ac.uk

## ABSTRACT

**Objectives** Blunt chest trauma (BCT) is characterised by forceful and non-penetrative impact to the chest region. Increased access to the internet has led to online healthcare resources becoming used by the public to educate themselves about medical conditions. This study aimed to determine whether online resources for BCT are at an appropriate readability level and visual appearance for the public.

**Design** We undertook a (1) a narrative overview assessment of the website; (2) a visual assessment of the identified website material content using an adapted framework of predetermined key criteria based on the Centers for Medicare and Medicaid Services toolkit and (3) a readability assessment using five readability scores and the Flesch reading ease score using Readable software.

**Data sources** Using a range of key search terms, we searched Google, Bing and Yahoo websites on 9 October 2023 for online resources about BCT.

**Results** We identified and assessed 85 websites. The median visual assessment score for the identified websites was 22, with a range of −14 to 37. The median readability score generated was 9 (14–15 years), with a range of 4.9–15.8. There was a significant association between the visual assessment and readability scores with a tendency for websites with lower readability scores having higher scores for the visual assessment (Spearman's r=−0.485; p<0.01). The median score for Flesch reading ease was 63.9 (plain English) with a range of 21.1–85.3.

**Conclusions** Although the readability levels and visual appearance were acceptable for the public for many websites, many of the resources had much higher readability scores than the recommended level (8–10) and visually were poor.

Better use of images would improve the appearance of websites further. Less medical terminology and shorter word and sentence length would also allow the public to comprehend the contained information more easily.

## INTRODUCTION

Blunt chest trauma (BCT) is the leading cause of death among young adults, aged 15–44, and is second only to head trauma as the most common cause of death for all age groups.[1] The UK's major trauma population is now reported to be more elderly, and the predominant mechanism is a fall from less than 2 m.[2]

Since the start of the century, internet usage across the globe has increased by 1355%, with more than 5 billion internet users in the world as of 2023.[3] This huge increase in access to the internet has led to people using it as a healthcare resource, with 61% of Americans having looked up health-related information.[4] A study found that 62% of people considered healthcare information on the internet to be 'excellent' or 'very good', and over half of people in the study felt they did not feel the need to share their findings with a doctor.[5] This suggests that many people see the internet as a valuable and accurate tool for healthcare. This number is likely to keep growing and this reliance on internet resources for all forms of healthcare means it is essential that information is accessible, accurate and readable.

Health literacy is defined as 'a person's ability to understand and use information to make decisions about their health and is believed to have a vital impact on public health due to access and use of health

services.[6 7] Between 43% and 61% of adults in the UK do not routinely understand health information,[8] with more than 7 million adults having 'very poor literacy skills'.[9] This means that they experience difficulties reading from unfamiliar sources. Similarly in the USA, the reading level of the average person is at an eighth-grade level (age 12–13 years), meaning that they cannot interpret and evaluate any information that requires inference.[10]

### STRENGTHS AND LIMITATIONS OF THIS STUDY

⇒ We undertook a rigorous assessment of identified websites containing information on blunt chest trauma.
⇒ We calculated extensive readability statistics for each website using five different readability scores and the Flesch reading ease score.
⇒ We supplemented the readability assessment with a visual assessment using defined criteria adapted from the Centers for Medicare and Medicaid Services toolkit.
⇒ We searched for relevant website materials using three search engines and restricted assessment to those websites identified on the first search page, therefore, findings are limited to resources identified using these search engines.

Low health literacy levels are a potential barrier to the use of online health resources. If the information is not able to be understood by a person without clinical knowledge, then this could hinder early diagnosis and treatment of potentially life-threatening conditions. For this reason, it is essential that online resources are written at a low readability level, meaning that they are simple to interpret and understand.

Previous studies have found that online resources for healthcare are at a high readability level and are of poor quality,[11–15] showing that they contain medical terminology and are not visually appealing. For BCT, a study assessing the readability of a trauma surgery website found that the readability level was too high, and therefore, difficult to comprehend.[16]

One of the authors (CB) has developed a predictive risk tool to help manage patients with BCT[17 18] with one of the management pathways being to be sent home. It is important that patients understand any materials they are given. It is likely that such patients will access more materials online and as such we thought it was important to assess such materials. Apart from the study assessing the readability of the trauma surgery website, there has been no previous research undertaken assessing the quality and readability of online resources for BCT. The aim of this study was, therefore, to carry out an assessment of the readability and visual appearance of online resources for BCT.

## METHODS
### Identifying online resources
To identify relevant websites we used the three most popular search engines in the UK: Google, Bing and Yahoo.[19] In order to capture the most relevant websites, we used a range of search terms. These were based on terms used in the existing literature. In order to capture more colloquial terms that would be familiar to patients, we also consulted with clinical colleagues who manage patients with chest trauma. Box 1 presents details of the full list of search terms used.

We identified all the websites that appeared on the first page of each search engine. We undertook: (1) a narrative overview assessment; (2) a readability assessment and (3) a visual assessment of each website.

In order to evaluate the accessibility of the identified materials in terms of readability and visual aesthetics of online resources for BCT, we aimed to identify relevant websites that contained information intended to educate the public. We, therefore, excluded: research publications;

| Box 1  List of search terms used to undertake searches in Google, Bing and Yahoo |
| --- |
| ⇒ blunt chest trauma |
| ⇒ blunt chest injury |
| ⇒ chest impact injury |
| ⇒ chest trauma |
| ⇒ thoracic contusion |
| ⇒ forceful chest injury |
| ⇒ chest injury |
| ⇒ broken ribs |
| ⇒ rib fracture |
| ⇒ chest and rib injury |
| ⇒ chest wall injury |
| ⇒ rib injuries |
| ⇒ bruised ribs |

books/chapters; clinical guidelines; teaching resources; clinical case studies; risk calculators; conference papers; newspaper articles; legal support services; referral information for clinicians; radio transcripts; sites solely for advertisement only and sites where contact information only was provided. We further excluded sites where the content was clearly not relevant to the condition; private websites with restricted access and sites with only video resources. Scientific papers, teaching resources and clinical guidelines are not resources developed specifically to educate and inform the public about BCT, while websites that require payment receive low traffic, with previous research finding that 80% of people who encounter a paywall when looking for health information will choose to search elsewhere.[3] We included Wikipedia, despite the fact that its content can be changed and hence may not contain accurate information.[20] We decided to include it as it is one of the most commonly accessed websites (currently seventh), and therefore, it is important to assess its quality, due to the high level of traffic it receives.[20 21]

### Narrative overview assessment
We first produced a narrative overview assessment of each of the identified websites. We produced a paragraph that summarised our initial impressions of the website outlining areas that were good or poor, prior to undertaking more detailed visual and readability assessments.

### Visual assessment
We used the 'Guidelines for effective writing',[22] which details key areas that may affect the readability of websites in order to visually assess each website. These included headings, content organisation and language use, text size and colour, use of white space, and illustrations (for full list, see online supplemental table S1).

We scored the visual assessment for each website. We compiled a list of 42 criteria for the assessment based on the 'Guidelines for effective writing'.[22] We selected these criteria as we judged them to be the most important elements when assessing the written content and visual appearance of the webpage. For each of these 42 criteria,

we assigned a score of +1 point if the criterion was achieved, 0 if the criterion was not applicable and −1 point if the criterion was not achieved. These points were added to give a maximum cumulative score of 42 for each website.

## Readability assessment

When assessing the readability of each website, we used the website Readable (https://readable.com) to generate several readability scores for each online BCT resource. To do this, we entered each page of text that was considered relevant to BCT into Readable. Readable then generated various readability statistics. All text that appeared on any relevant page was entered into readable, including headings and lists. However, all images, graphs and navigation areas were removed for the purposes of calculating readability. These were, however, considered as part of the visual assessment of the websites.

When analysing the content of our chosen websites, we assessed all the relevant pages, up to a maximum of 10 pages. We only included pages that specifically discussed BCT or related injuries and disregarded any that contained other healthcare information or contact details.

We used five different readability formulae to give a wide evaluation of each website. There is currently no consensus regarding which readability formula is the most appropriate for assessing patient materials and it is, therefore, recommended that more than one formula is used to assess such materials[23] Each formula assesses the text in a different way and includes items such as words, characters and syllables.[12] Employing a range of scores, therefore, improves the validity of the results.[23] We used the readability formulae: Flesch-Kincaid Grade Index,[24] Coleman-Liau Index,[25] Simplified Measure of Gobbledygook Index,[26] Gunning-Fog Index[27] and the Automated Readability Index.[28] We chose these readability formulae as they have previously been used to assess readability within the medical field.[15 16 23 29–31]

In addition to calculating the readability formulae, we also calculated the percentage of the population the text could be read by and the Flesch Reading Ease (FRE) score.[32] The FRE score is the earliest of the commonly used tools to assess readability[23 31] and gives a score on a scale ranging from 0 to 100. A score of 0 is classified as being unreadable and that of 100 the most readable.[31 32] It is based on the average number of syllables per word and the average number of words per sentence. Content with a score of 70 is easy to read for most of the population. Text with shorter sentences and simpler words will have a higher score than text with longer sentences and more complex words.[32] We used the FRE score to provide a summary score of the accessibility of the text within the identified websites.

Readable generated scores as a school-grade level, which was then translated into a corresponding age. This is the age that could be expected to comprehend the piece of writing. Various healthcare organisations have recommended that readability levels should be between sixth and no more than eighth grade (age 11–14 years).[33–35]

## Changes made to original protocol

Following peer review feedback from reviewers, the original protocol was expanded to increase the breadth of websites reviewed. The search terms used were expanded to include terms that were more likely to be understood and used by patients. Our original protocol only used the terms: 'blunt chest trauma', 'blunt chest injury', 'chest impact injury', 'chest trauma', 'thoracic contusion' and 'forceful chest injury'. We also assessed all relevant websites identified on the first page of each search engine rather than the top 10. In our original protocol, we undertook a limited visual assessment using only 10 criteria. We modified this to undertake a comprehensive assessment using 42 criteria following review.

## Statistical analysis

The readability statistics were generated for each page of the website used. A median (range) readability score was calculated based on the five readability formulae. Where more than one page was assessed for a website, scores were aggregated to give a median score for each formula that was representative of the entire website.

We compared readability scores with the cumulative score from the visual assessment to determine if there was any correlation between readability and visual appearance, using non-parametric Spearman correlation. A $p<0.05$ was regarded as being statistically significant.[36]

## Patient and public involvement

Due to the limited time available to complete this project and lack of funding, it was not possible to involve patients or the public in the design, or conduct, or reporting, or dissemination plans of our research.

## RESULTS
### Identification of online resources

Using the search terms across the first page of the three search engines, we identified a total of 258 websites. We identified a large amount of repetition of websites and a large number of websites were excluded based on applying our predefined search criteria (see figure 1). There were a large number of scientific papers identified, especially when using more clinical search terms such as 'thoracic contusion'. When using search terms that were less scientific such as 'bruised' or 'broken ribs', more online resources appeared that were more suitable for use by the public. After applying exclusions, we identified a total of 85 websites for further assessment.

### Narrative overview assessment

Online supplemental table S2 details the online resources identified, with a brief written assessment of their content. Many of the resources that we identified were not solely based on BCT. Instead, they contained

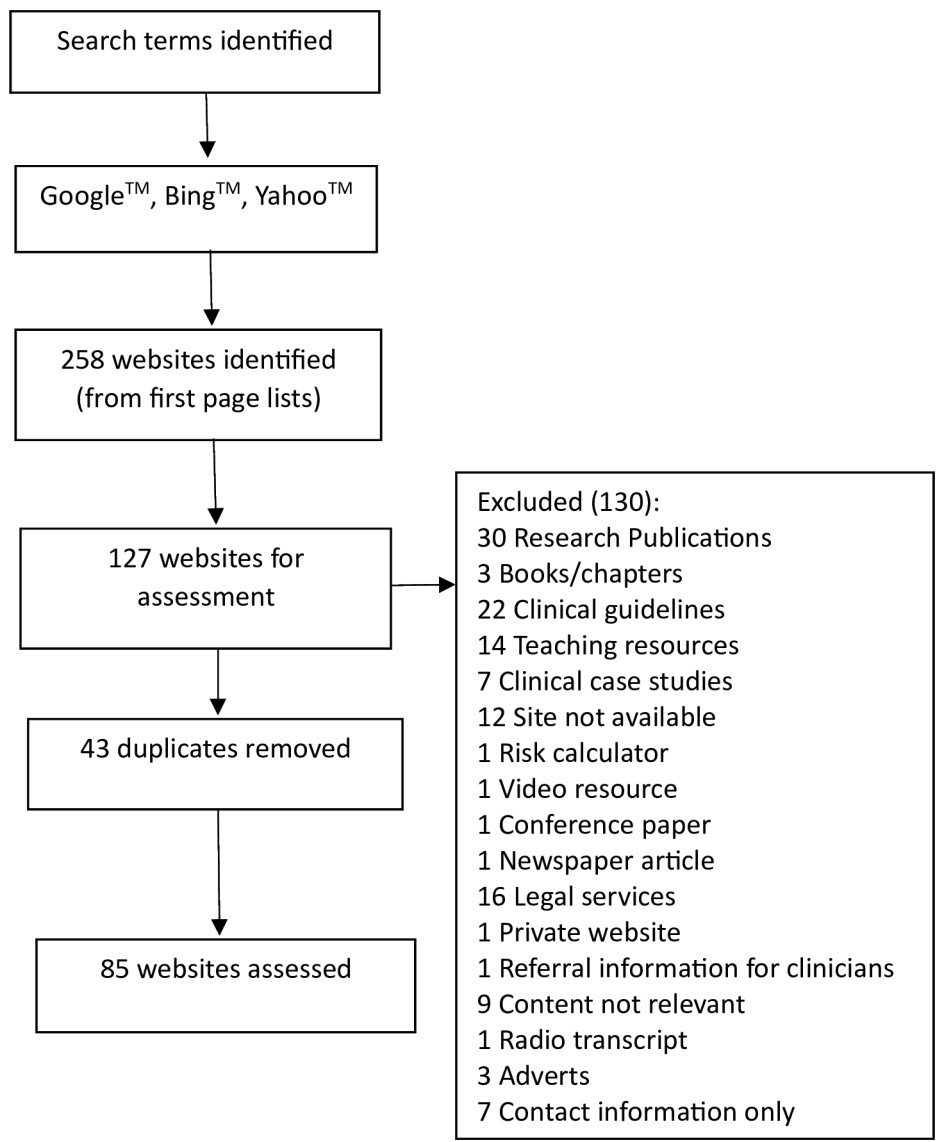

**Figure 1** Flow diagram illustrating number of websites identified and reasons for exclusions.

sections that were relevant to BCT as well as information on either other types of chest injury, or trauma injuries affecting the head or abdomen. Similarly, some of the resources were focused on types of injury caused by BCT, such as pneumothorax, flail chest and rib fractures. Most of the information identified was contained within a single webpage. There was limited use of photographs and illustrations that could have potentially improved the visual aesthetics of the website and understanding of the information. Many of the websites linked to simple patient information leaflets that focused on advice for patients to help manage chest injuries/rib fractures/bruising with signposting for further help should they need it. Many sites linked to the National Health Service (NHS) (eg, general practice surgeries and secondary care hospitals) and simply replicated the information from the NHS site and added their own branding. Overall, there was very limited, patient-focused information identified for BCT.

### Visual assessment

Details of the visual assessment for each of the 85 online resources are shown in online supplemental table S1. There was a large amount of variability across the websites. The median visual assessment score was 22 with a range from −14 to 37 out of a maximum possible score of 42. More than 50% of the websites achieved a score of 22 or less (45/85 websites; 52.9%). This means that most websites either did not consider or scored negatively on almost half of the criteria identified as important based on the Centers for Medicare and Medicaid Services Toolkit for Making Written Material Clear and Effective.[22] This indicates that most websites could do more to make their websites more visually appealing.

The lowest scoring resource was the ISK Institute website, with an overall score of −14 from a possible 42. Other poorly scoring websites with a score of less than 10 were: World Rugby Passport (−3); GP notebook (6); Wikipedia (6); Physiocheck (7); Farrell Physiotherapy (8);

Orthopaedics and Trauma London (8) and DynaMed (8). Poorly scoring websites appeared cluttered with too much complex information and extensive use of medical terminology, poor use of colouring and contrast, distracting adverts and limited or inappropriate use of illustrations. Many of these websites seem to be advertising services and, therefore, appear less credible.

The highest scoring website was the Royal Devon University Healthcare NHS Foundation Trust & Northern Devon Healthcare NHS Trust with 37. There were five other websites that scored highly on their visual assessment with a score of 32 or more (Cleveland Clinic (32); Dr Gallagher and Partners (32); Bradford Teaching Hospitals NHS Foundation Trust (34); healthinfo.org. nz (34); Agency for Clinical Innovation (36)). Highly scoring websites had a clean and uncluttered appearance with lots of white space, sparing use of colour and good contrast of colours, easy to understand information with limited use of medical terminology, no adverts and appropriate use of illustrations that supported the text content. These websites largely were associated with credible organisations such as the NHS or other government bodies, which helps to reassure the public that the information is more credible.

### Readability assessment

Online supplemental table S3 provides detailed information on the readability assessment. Most websites only had one page of information on which to undertake a readability assessment.

The median readability score across all websites was 9, which equates to a reading age of 14–15 years of age. The range of scores was from 4.9 (10–11 years of age) to 15.8 (18+ years of age or university level). Aiming for a readability grade of a maximum of 8,[33–35] only 30 of the 85 websites identified (35.3%) were at an appropriate reading level for the general public.

The median FRE score was 63.9 (Plain English) with a range between 21.1 (very difficult) for the UpToDate website, to 85.3 (easy) for the Doncaster and Bassetlaw Teaching Hospitals website. This is understandable given that the UpToDate website uses more medical terminology and seems to be directed at clinicians, whereas the Doncaster and Bassetlaw Teaching Hospitals website is a resource created for patients/the public. Only two additional websites were also classified as easy to read with an FRE score of more than 80 (NHS (81.1); My Health Alberta (81.7)). Aiming for an FRE score of greater than 70[32] only 29/85 (34.1%) were accessible for most of the population.

In terms of reach and the percentage of the addressable audience that each website resource was readable to, scores ranged from 50% for the website UpToDate, to 100%, which was scored by more over 60% (52/85) of the websites.

### Comparison between visual assessment and readability scores

Figure 2 shows the median readability score and the visual assessment score for each online resource. There was a large amount of variation between the websites for both categories. Some websites performed well in the readability assessment but badly in the visual assessment and vice versa. There was a statistically significant association between the visual assessment and readability scores with a tendency for websites with lower readability scores

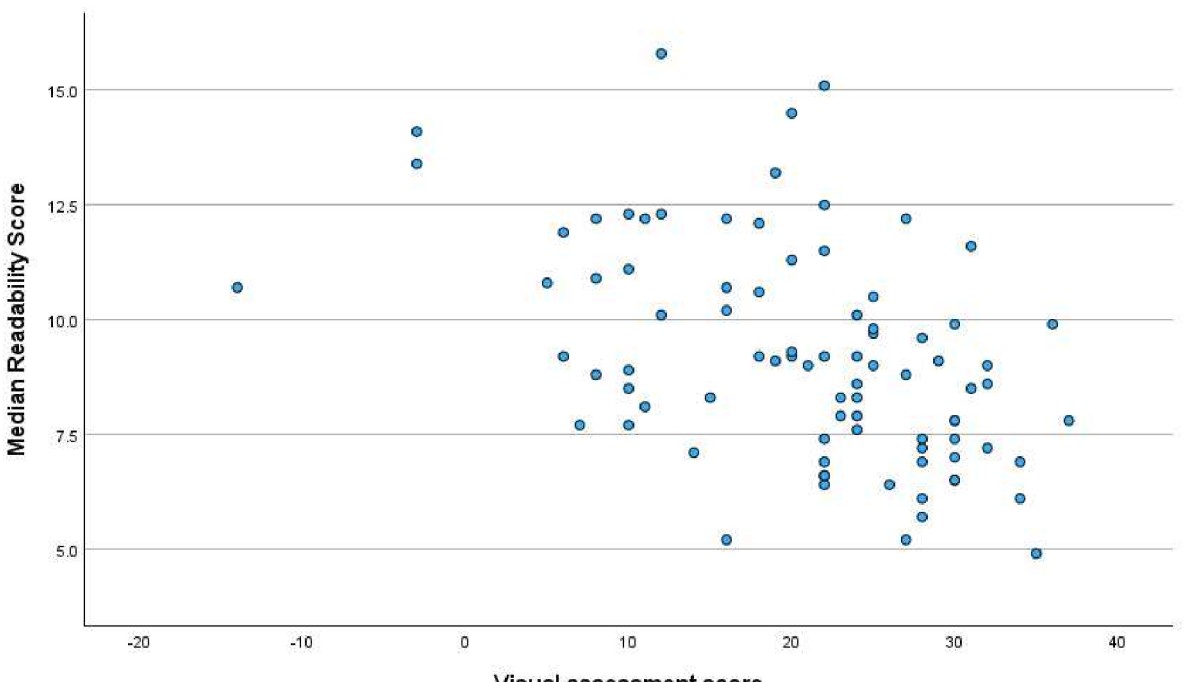

**Figure 2** Median readability score versus visual assessment score for the 85 assessed websites. Spearman's r=−0.485; p<0.01.

(ie, easier to read) having higher (ie, better) scores for the visual assessment (Spearman's r=−0.485; p<0.01). This suggests that there is a correlation between websites being difficult to read and being visually unappealing. A low readability score and a high visual assessment score are most desirable.

## DISCUSSION

Overall, we found that online resources for BCT were generally visually appealing, but the content was written at a level too high for the public to understand. This shows resources available for public access are not providing accessible information. This may lead to a misunderstanding of the information being provided and consequentially could result in slower diagnosis and greater risk of hospitalisation.

To be deemed readable for the general public, a readability grade score of between 6 and 8 should be achieved, which equates to a reading age of between 11 and 14 years. Eighth grade is the average reading age of an American adult,[10] so if the readability score of online resources are any higher than this, the information contained in the online resource will be inaccessible to more than half of the general public. We calculated a median readability score for the identified BCT resources of 9 (age 14–15 years), which is higher than the recommended score of 8. Only 35% of the websites had a readability of less than 8,[34 35] which suggests that online resources for BCT are not suitable for educating the public. This is supported by the fact that only three websites were classified as easy to read with a score of more than 80 when considering the FRE score,[32] with only 34% being accessible for most of the population.

Our findings concur with previous research assessing readability of online resources for other health conditions, such as phenylketonuria, fibroadenoma, otolaryngology and parathyroidectomy, which all identified that readability levels were too high.[11–14] This suggests that readability issues are common in online health resources, which could be a reason for up to 61% of UK adults not routinely understanding health information.[8]

These findings suggest that it is essential for those writing online resources for BCT to consider their target audience, and to ensure that the content of the website is accessible and understandable for the public. Possible considerations to improve public accessibility include the use of shorter words and sentences and the use of less complex words and medical terminology throughout the text. Previous studies have demonstrated that shortening sentences to less than 15 words led to an improvement in readability.[37] In certain scenarios where using more complex terms is necessary, brief explanations or definitions can also be helpful to the reader. All these factors can contribute to the production of more readable online resources for BCT, leading to increased levels of education for the public regarding their health.

For the visual assessment of BCT resources, a median score of 22/42 was recorded and a large number of the websites were generally well laid out and visually appealing. However, with no website scoring 42 and with more than 50% of websites scoring less than 22 this indicates that there are still ways in which the visual appearance of these websites can be improved.

It is important for the online resource to be appealing as this is the first thing a user will see and may impact whether they choose to continue to read the information present on the webpage.[38–40]

Even if a resource for BCT contains accurate and helpful information written at an appropriate readability level for a member of the public, not having a visually appealing page may cause the reader to look elsewhere.

In designing a website for BCT education, guidance can be taken from the 'guidelines for effective writing' from the Centers for Medicare and Medicaid Services.[22] This will provide guidance on how best to structure text, with the use of paragraphing and headings essential to break up text into more manageable sections for readers. Using bullet points to highlight key information is also beneficial to the reader, being clear and concise.[41] This is more appealing than having to search for information in a large block of text.

It is important to consider making information accessible to people who may have poor eyesight or have problems with reading. The use of images and audio aids can be effective in communicating BCT information to these audiences. The online resources assessed in this study demonstrated limited use of images or diagrams, suggesting that less importance is placed on visual aids compared with written information. Previous research has found that the use of audio and visual aids has a significant impact on learning.[42] This further highlights the importance of using these aids in future online resources for BCT.

Many websites only partially discussed BCT or focused more on injuries that could be caused by BCT. In addition, several websites were aimed at audiences with more advanced scientific knowledge, however, this was not clearly identifiable when the website was first accessed. These issues could both lead to confusion for the reader and dissuade them from looking for online information for BCT in the future. Therefore, it is important for future online resources to not only be visually appealing and written at an appropriate level, but to be clearly identifiable as resources intended for use by the public.

There are some limitations to this study. We employed a range of search terms, including many that would be more familiar to the public. However, a larger range of search terms could have been used, which may have identified more online resources. A larger number of resources could also have been assessed. However, since the three search engines we chose to use receive over 95% of traffic, and that 95% of people do not go past the first page of results[43] we believe that we identified the websites that would receive almost all the traffic. We also identified

a lot of replication in the 258 identified websites within the first page search, which is likely to increase further past the first page of searches. Apart from assessing some criteria relating to content accuracy as part of the visual assessment (such as author, use of references and website updates), we did not undertake a full assessment of the accuracy of the website content. The purpose of the study was to undertake an initial assessment of the readability and visual appearance of online materials for access by members of the public/patients. It was beyond the scope of the study to assess website content and further work is, therefore, needed to assess the accuracy of the materials.

When assessing readability and visual quality of the chosen resources, we only used five readability formulae, all of which assessed the readability based on sentence and word length and complexity. This did not consider other factors such as tables and diagrams that could also affect how easily a website can be read. Even though these were considered in the visual assessment, evaluating tables and figures only formed a small part of the visual assessment. A specific examination of visual and audio aids would be useful for future research to consider when assessing the quality of online resources.

Some websites with only a small amount of information may demonstrate an overly high readability score, and not be as representative of the overall readability of the website as those in which multiple pages of information were assessed. However, if someone is searching for BCT information, it is less important to them how readable other sections of the websites are.

Further research is necessary, assessing specific aspects of visual appearance such as images in detail. Deeper analysis of the accuracy of the scientific content of the websites could also be useful, and other online resources such as videos or audio content should be considered.

## CONCLUSIONS

We found that online resources for BCT were written at a level too advanced for use by the public, with a reading age greater than recommended. The visual appearance of these resources was generally at a level acceptable to the public. BCT online resources could, however, be made more accessible and improved for public use by reducing the reading age of the textual content, and by considering additional criteria to improve visual aesthetics, such as the use of images.

**Contributors** HAH designed the original study, quality assured the data collection and analysis, provided supervisory support and drafted the manuscript. MC adapted the study design, undertook the data searches, analysed the data, drafted data outputs and reviewed the final manuscript. CB provided input into study design, provided clinical advisory support to the project, quality assured the outputs and reviewed the final manuscript. HAH is the guarantor for the study.

**Funding** The authors have not declared a specific grant for this research from any funding agency in the public, commercial or not-for-profit sectors.

**Competing interests** None declared.

**Patient and public involvement** Patients and/or the public were not involved in the design, or conduct, or reporting, or dissemination plans of this research.

**Patient consent for publication** Not applicable.

**Provenance and peer review** Not commissioned; externally peer reviewed.

**Data availability statement** Data are available in a public, open access repository. Data are available upon reasonable request from the corresponding author.

**ORCID iDs**
Hayley Anne Hutchings http://orcid.org/0000-0003-4155-1741
Ceri Battle http://orcid.org/0000-0002-7503-1931

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
