## [Reviewer comments · BMJ Open]

ARTICLE DETAILS

TITLE (PROVISIONAL)	A study to evaluate the readability and visual appearance of online resources for blunt chest trauma: an evaluation of online resources using mixed methods
AUTHORS	Hutchings, Hayley; Cochrane, Max; Battle, Ceri

VERSION 1 – REVIEW

REVIEWER	Professor Samy Azer King Saud University College of Medicine, Medical Education
REVIEW RETURNED	02-Sep-2023

GENERAL COMMENTS	A study to evaluate the readability and visual appearance of online resources for blunt chest trauma BMJ Open Thank you for inviting me to review the above-titled manuscript. The topic is exciting and matches the scope of the journal. However, there are problems in the manuscript. Abstract: 1) Content accuracy is missing. It should be included. 2) State the search words used. Also, the date of the search, and the inclusion and exclusion criteria. 3) Only 30 websites- on what basis? Is this after excluding? You should have included a PRISMA figure in the manuscript showing these details. 4) The visual assessment score system should be included in the appendix. 5) You need a control for the visual assessment score. It should be compared with another score system included in the literature. Your conclusion, "The visual appearance of websites was generally good," is meaningless without control. 6) What is Fles Reading Ease? We usually use the Flesch-Kincaid Grade Index and the Coleman-Liau Index. 7) The results are shallow and there should be parameters the researchers explored for each website (see later). 8) The results of the research should show something deeper than what the public will do on using the visual assessment score. We need to know- topics addressed, the strengths, the limitations of each website, the creator, the date created, updates status, content accuracy, links with other resources, videos, graphs, animations, references, readability level etc. Results should show these findings in figures- mean \pmSD, median IQR, correlation, p-value, etc. 9) conclusion as stated earlier, not based on strong evidence. 10) Images and diagrams are mentioned in the conclusion but not in the results. Strengths- 1) Should be reduced to 3 points. 2) Point 2 is not true- content accuracy was not measured and no control to compare. 3) readability- Usually, we use two methods for specific purposes; using 5 or 8 without a purpose that can help the research question is redundant and can distract the readers and raise hundreds of
--

	questions (points 3 and 4). It would help if you compare your findings against another tool (control). Introduction- 1) What triggered the study? 2) What is your research question? 3) Content accuracy should be included, 4) Justify why the visual assessment score system was used, you need a control to compare against. Methods- 1) State a reference for the search words used. 2) Clearly state the inclusion and exclusion criteria. 3) Justify why the visual assessment score system was used, you need a control to compare against. 4) add a reference for the readability method used. 5) Usually, we use two methods for specific purposes; using 5 or 8 without a purpose that can help the research question is redundant and can distract the readers and raise hundreds of questions. Omit other redundant readability methods used as they were of no value in your discussion (see other comments). 6) Content accuracy should be measured. 7) State the parameters examined for each website (see earlier comments). 8) Statistical assessment- add a reference. What tests did you use? Results- 1) we do not have a control, and no assessment of content accuracy, and no parameters examined for each website. 2) A PRISMA Figure is needed for the search and the websites finally included. 3) Needs subtitles. 4) Tables should be reviewed and amended. Discussion- 1) Shallow should be rewritten after the suggested changes have been made. 2) Discuss proper studies published in BMJopen and other high-impact journals. 3) What are the limitations of the study? Conclusion- 1) What does "the visual appearances of these website resources was generally good" mean? You need a control References should be improved see for example - doi: 10.1136/bmjopen-2015-008187; doi: 10.1371/journal.pone.0228786; DOI: 10.1097/MEG.0000000000000003
--	--

REVIEWER	Jared Wohlgemut Queen Mary University of London, Centre for Trauma Sciences, Blizard Institute
REVIEW RETURNED	11-Sep-2023

GENERAL COMMENTS	Abstract Objectives- Delete a space before "This" on the last sentence. Design- the words "based on" are used twice in one sentence- suggest replacing the second one with "using the", or equivalent. Data Sources- good Results- called qualitative, but most results are quantitative. Conclusions- suggest changing the words "good" to "acceptable for the public" or "appropriate for the public". Introduction 2nd paragraph, 2nd sentence, should read "health-related". 3rd paragraph, 4th sentence, I would re-word to "the reading level of the average person is at the 8th grade level", because the way it is written sounds overly negative. Methods
---

	Identifying online resources- first sentence- delete the word "top"- it is redundant. Regarding search terms used, perhaps more colloquial language would be more useful in this methodology, such as "sore chest", "bruised rib", "have I broken my ribs?", or "sore to breathe", for example. How would the result change when the search terms used reflect the language one would expect would be used when the public 'Google' their symptoms? Because at the moment, the study uses medical terms for search terms, then judges websites for being too medically-focused (using jargon). This method seems incongruous. Identifying online resources- last sentence- do you have a reference that Wikipedia receives a high level of traffic? Visual assessment- second sentence- add "of" between "readability" and "websites". Visual assessment- third sentence- instead of "amongst others", please list all key areas that may affect the readability of websites. Visual assessment- second paragraph, second sentence- delete a space between "19" and "." Readability assessment- Much of your methodology (in paragraph 3 and 4) references a proprietary website (Readable), rather than published research. Please find alternative references for "have become an important tool in medicine for developing resources for the public" and "the most accurate way to score readability", and that the UK general public school grade 8, reading age of 12-13 years, equates to 85% of the population. The Flesch Reading Ease score is not adequately described or justified as part of the methods. No mention of an Equator guideline used. It was described as qualitative research, though it is better described as mixed methods given the quantitative results displayed. Might SRQR or COREQ be applicable to this paper? Or, given it was an evaluation of currently available formats compared to a standard reading and visual level of the public, might this be considered a service evaluation, where SQUIRE would be appropriate? https://www.equator-network.org/reporting-guidelines/srqr/ https://www.equator-network.org/reporting-guidelines/coreq/ https://www.equator-network.org/reporting-guidelines/squire/ Results Visual assessment- fourth paragraph, second sentence, suggest changing to "The text was characterised by long paragraphs..." Readability assessment- My understanding is that UpToDate is primarily for clinicians, in providing clinical decision support, so it is understandable that it achieves less readability than a government health website aimed at patients (MyHealth.Alberta). Discussion First paragraph, first sentence- suggest finishing the sentence with "at a level too high for the public to understand" or equivalent. First paragraph, third sentence- typo: change "is" to "a" misunderstanding... Generally, numbers less than 10 should be written ("one"), which applies in several instances in the Discussion. Sixth paragraph- Do you have a reference for this psychological behaviour "It is important for the online resource to be appealing as this is the first thing a user will see and may impact whether they choose to continue to read the information present on the webpage" Ninth paragraph- Suggest removing "at first glance", and replacing with "initially" or "when the website was first accessed".
--	---

	Tenth paragraph- I don't agree with the statement "it is also essential for these websites to be focused entirely on BCT", because patients are often multiply injured (not just thorax). Should they really be expected to search for a different dedicated website for each injury they have suffered? I think this is unrealistic, and may add extra effort and stress for patients seeking health information, if they have to identify multiple websites for advice. Twelfth paragraph- "this was only to a brief degree" is vague and awkward language. Suggest changing the sentence to something like "Evaluating tables and diagrams formed a small part of the visual assessment." Conclusions Second sentence- suggest changing "good" to "appropriate" or "acceptable to the public" Last sentence is weak- suggest make a bolder statement justified by findings, such as "BCT online resources could be more effective for public use with reducing the reading age of textual content and improving the visual appearance." or similar. Overall A simple, clear study, in which the motivation for the study was clear, the aim achievable, and the methods mostly appropriate. However, the search terms could have reflected more colloquial language (similar to public language), which potentially biased their results towards overly medical websites (websites that were aimed at clinicians). Second, there was no use of an Equator guideline/checklist. Third, there were statements that were inappropriately or inadequately referenced. Fourth, there were some minor typos. Therefore I've suggested it is reconsidered after revisions.
--	---

VERSION 1 – AUTHOR RESPONSE

Reviewer: 1

Prof. Professor Samy Azer, King Saud University College of Medicine, The University of Sydney
Faculty of Medicine and Health

Comments to the Author:

A study to evaluate the readability and visual appearance of online resources for blunt chest trauma
BMJ Open

Thank you for inviting me to review the above-titled manuscript. The topic is exciting and matches the scope of the journal. However, there are problems in the manuscript.

Abstract: 1) Content accuracy is missing. It should be included.

Response: The purpose of the study was to undertake an initial assessment of the readability and visual appearance of online materials for access by members of the public/patients. We agree that it is important to assess the accuracy of the materials, but it was beyond the scope of the study. We have acknowledged this as a limitation in our discussion section.

We have however now undertaken a more extensive evaluation of the online materials (please see Supplementary information S1) that includes assessment of whether appropriate references are included and whether administrative information (e.g. authors, last review date) is included on the website.

2) State the search words used. Also, the date of the search, and the inclusion and exclusion criteria.

Response: We have now included the dates of search to the abstract. We have provided fuller details about the search terms and the inclusion and exclusion criteria to the methods section only due to word count restrictions for the abstract.

3) Only 30 websites- on what basis? Is this after excluding? You should have included a PRISMA figure in the manuscript showing these details.

Response: We made a pragmatic decision to include all websites identified on the first page of the search engines. This was based on previous research that states that people rarely go beyond the first page of a search. We had already identify substantial overlap across sites. We have added more text to clarify this in the methods section.

Based on comments from the second reviewer we have extended the searches utilising search terms that may be more familiar to a lay person. This resulted in a total of 85 websites.

We have now included a flow diagram illustrating the number of websites identified and the reasons for exclusion.

4) The visual assessment score system should be included in the appendix.

Response: Thank you. We agree that given the scale of the table, that this assessment should now be an appendix/supplementary table.

5) You need a control for the visual assessment score. It should be compared with another score system included in the literature. Your conclusion, "The visual appearance of websites was generally good," is meaningless without control.

Response: We have now undertaken a more extensive visual assessment, which was used alongside and compared with the readability assessment. We have presented the results in terms of proportion of websites achieving various cut-offs and have removed the term 'generally good'.

6) What is Fles Reading Ease? We usually use the Flesch-Kincaid Grade Index and the Coleman-Liau Index.

Response: We apologise for not including more information about the Flesch Reading Ease score. We have now included more detail in the methods and have further justified why we used it.

7) The results are shallow and there should be parameters the researchers explored for each website (see later).

Response: The aim of the study was to provide an initial assessment of online resources for blunt chest trauma. We have now reviewed 85 websites and provide a narrative summary, an extensive visual assessment of content using 42 separate criteria (based on the Centers for Medicare and Medicaid Services Toolkit for Making Written Material Clear and Effective) and a comprehensive Readability assessment utilising 5 different readability scores, the Flesh Reading Ease score and an assessment of what proportion of the public the information is readable to. We do not believe that this is shallow. We recognise that there are limitations, and that assessment of clinical content is important, but this was beyond the scope of the study. In addition, we have acknowledged this in our discussion.

8) The results of the research should show something deeper than what the public will do on using the visual assessment score. We need to know- topics addressed, the strengths, the limitations of each website, the creator, the date created, updates status, content accuracy, links with other resources, videos, graphs, animations, references, readability level etc. Results should show these findings in figures- mean \pm SD, median IQR, correlation, p-value, etc. 9) conclusion as stated earlier, not based on strong evidence. 10) Images and diagrams are mentioned in the conclusion but not in the results.

Response: We have now undertaken a more comprehensive assessment of the websites. Our summary narrative overview of the website indicates whether illustrations/figures were included. Our visual assessment has also now been extended to include a more comprehensive assessment of 42 criteria that relate to the website content and presentation. This includes, for example, whether visuals/illustrations were included and administrative information about the website, for example date created, last updated date and reference list.

We have updated the relevant sections in the manuscript to describe the assessments and the findings. We have changed the results to include appropriate statistics. We have modified the conclusion.

Strengths- 1) Should be reduced to 3 points. 2) Point 2 is not true- content accuracy was not measured and no control to compare. 3) readability- Usually, we use two methods for specific purposes; using 5 or 8 without a purpose that can help the research question is redundant and can distract the readers and raise hundreds of questions (points 3 and 4). It would help if you compare your findings against another tool (control).

Response: 1)2) We have reduced the strengths section and made reference to content assessment not being measured as a limitation. 3) We have further justified why we have used five different readability formulae. As we have now undertaken a more extensive visual assessment, we believe that comparison against another tool is beyond the scope of the study.

Introduction- 1) What triggered the study? 2) What is your research question? 3) Content accuracy should be included, 4) Justify why the visual assessment score system was used, you need a control to compare against.

Response: 1)2) We have added to the introduction details of why we undertook the study and our research question. 4) We have provided further justification for using the visual assessment tool. 3) We have specified why we did not undertake a content assessment.

Methods- 1) State a reference for the search words used. 2) Clearly state the inclusion and exclusion criteria. 3) Justify why the visual assessment score system was used, you need a control to compare against. 4) add a reference for the readability method used. 5) Usually, we use two methods for specific purposes; using 5 or 8 without a purpose that can help the research question is redundant and can distract the readers and raise hundreds of questions. Omit other redundant readability methods used as they were of no value in your discussion (see other comments). 6) Content accuracy should be measured. 7) State the parameters examined for each website (see earlier comments). 8) Statistical assessment- add a reference. What tests did you use?

Response: 1) We have added further details about search terms and why they were used. 2) We have stated our inclusion and exclusion criteria. 3)4)5) We have justified why the visual assessment score was used and for the choice of readability formulae. 6) We have discussed why content assessment was not done. 7) We have added more detail regarding the parameters examined for each website. 8) We have added more detail regarding statistical assessments.

Results- 1) we do not have a control, and no assessment of content accuracy, and no parameters examined for each website. 2) A PRISMA Figure is needed for the search and the websites finally included. 3) Needs subtitles. 4) Tables should be reviewed and amended.

Response: 1) We have now undertaken a more extensive visual assessment of website that includes parameters of content accuracy. We have further elaborated on this issue throughout the manuscript. 2) We have now included a flow diagram to illustrate the searches, inclusions and exclusions. 3) We have added subtitles. 4) We have reviewed tables and amended appropriately.

Discussion- 1) Shallow should be rewritten after the suggested changes have been made. 2) Discuss proper studies published in BMJopen and other high-impact journals. 3) What are the limitations of the study?

Response: We have revised the discussion based on the edits made throughout the manuscript and included additional references. We have elaborated on limitations.

Conclusion- 1) What does "the visual appearances of these website resources was generally good" mean? You need a control

Response: We have rewritten the conclusion based on the comments made.

References should be improved see for example - doi: 10.1136/bmjopen-2015-008187; doi: 10.1371/journal.pone.0228786; DOI: 10.1097/MEG.0000000000000003

Response: We have checked the references can confirm that all details are correct.

Reviewer: 2

Dr. Jared Wohlgemut, Queen Mary University of London

Comments to the Author:

Abstract

Objectives- Delete a space before "This" on the last sentence.

Design- the words "based on" are used twice in one sentence- suggest replacing the second one with "using the", or equivalent.

Data Sources- good

Results- called qualitative, but most results are quantitative.

Conclusions- suggest changing the words "good" to "acceptable for the public" or "appropriate for the public".

Response: Thank you for these helpful comments. We have updated the Abstract accordingly.

Introduction

2nd paragraph, 2nd sentence, should read "health-related".

3rd paragraph, 4th sentence, I would re-word to "the reading level of the average person is at the 8th grade level", because the way it is written sounds overly negative.

Response: Thank you. We have updated the Introduction accordingly.

Methods

Identifying online resources- first sentence- delete the word "top"- it is redundant.

Response: This has now been corrected.

Regarding search terms used, perhaps more colloquial language would be more useful in this methodology, such as "sore chest", "bruised rib", "have I broken my ribs?", or "sore to breathe", for example. How would the result change when the search terms used reflect the language one would expect would be used when the public 'Google' their symptoms? Because at the moment, the study uses medical terms for search terms, then judges websites for being too medically-focused (using jargon). This method seems incongruous.

Response: Thank you for this helpful suggestion. We have now extended the searches to include more colloquial language that would be more familiar to the lay person.

Identifying online resources- last sentence- do you have a reference that Wikipedia receives a high level of traffic?

Response: We have now added a reference here.

Visual assessment- second sentence- add "of" between "readability" and "websites".

Visual assessment- third sentence- instead of "amongst others", please list all key areas that may affect the readability of websites.

Visual assessment- second paragraph, second sentence- delete a space between "19" and ".".

Response: These have now been corrected.

Readability assessment- Much of your methodology (in paragraph 3 and 4) references a proprietary website (Readable), rather than published research. Please find alternative references for "have become an important tool in medicine for developing resources for the public" and "the most accurate way to score readability", and that the UK general public school grade 8, reading age of 12-13 years, equates to 85% of the population.

The Flesch Reading Ease score is not adequately described or justified as part of the methods.

Response: Thank you for these helpful comments we have now added in appropriate references and provided more information for clarification.

No mention of an Equator guideline used. It was described as qualitative research, though it is better described as mixed methods given the quantitative results displayed. Might SRQR or COREQ be applicable to this paper? Or, given it was an evaluation of currently available formats compared to a

standard reading and visual level of the public, might this be considered a service evaluation, where SQUIRE would be appropriate?

<https://www.equator-network.org/reporting-guidelines/srqr/>

<https://www.equator-network.org/reporting-guidelines/coreq/>

<https://www.equator-network.org/reporting-guidelines/squire/>

Response: We have explored possible guidelines and believe none are appropriate for the methods we have used. We agree that the use of the word qualitative was not appropriate when describing our initial overview of the websites and have now changed this to narrative overview assessment.

Results

Visual assessment- fourth paragraph, second sentence, suggest changing to "The text was characterised by long paragraphs..."

Response: This has now been corrected.

Readability assessment- My understanding is that UpToDate is primarily for clinicians, in providing clinical decision support, so it is understandable that it achieves less readability than a government health website aimed at patients (MyHealth.Alberta).

Response: We have now added further discussion around the nature of the websites and how it may impact on readability.

Discussion

First paragraph, first sentence- suggest finishing the sentence with "at a level too high for the public to understand" or equivalent.

First paragraph, third sentence- typo: change "is" to "a" misunderstanding...

Generally, numbers less than 10 should be written ("one"), which applies in several instances in the Discussion.

Response: These have now been corrected.

Sixth paragraph- Do you have a reference for this psychological behaviour "It is important for the online resource to be appealing as this is the first thing a user will see and may impact whether they choose to continue to read the information present on the webpage"

Response: We have now added an appropriate reference.

Ninth paragraph- Suggest removing "at first glance", and replacing with "initially" or "when the website was first accessed".

Response: This has now been corrected.

Tenth paragraph- I don't agree with the statement "it is also essential for these websites to be focused entirely on BCT", because patients are often multiply injured (not just thorax). Should they really be expected to search for a different dedicated website for each injury they have suffered? I think this is unrealistic, and may add extra effort and stress for patients seeking health information, if they have to identify multiple websites for advice.

Response: Thank you for this observation. We have now deleted this statement.

Twelfth paragraph- "this was only to a brief degree" is vague and awkward language. Suggest changing the sentence to something like "Evaluating tables and diagrams formed a small part of the visual assessment."

Response: This has now been corrected.

Conclusions

Second sentence- suggest changing "good" to "appropriate" or "acceptable to the public"

Last sentence is weak- suggest make a bolder statement justified by findings, such as "BCT online resources could be more effective for public use with reducing the reading age of textual content and improving the visual appearance." or similar.

Response: These have now been corrected.

Overall

A simple, clear study, in which the motivation for the study was clear, the aim achievable, and the methods mostly appropriate. However, the search terms could have reflected more colloquial language (similar to public language), which potentially biased their results towards overly medical websites (websites that were aimed at clinicians). Second, there was no use of an Equator guideline/checklist. Third, there were statements that were inappropriately or inadequately referenced. Fourth, there were some minor typos. Therefore I've suggested it is reconsidered after revisions.

Response: Thank you for this helpful feedback. We have now amended the manuscript accordingly.